# Molecular Testing for the Diagnosis of Bacterial Vaginosis

**DOI:** 10.3390/ijms25010449

**Published:** 2023-12-28

**Authors:** Alevtina M. Savicheva

**Affiliations:** D.O. Ott Research Institute of Obstetrics, Gynecology and Reproductive Medicine, St. Petersburg 199034, Russia; savitcheva@mail.ru

**Keywords:** bacterial vaginosis, real-time PCR, vaginal microbiota, fluorescence in situ hybridization, diagnosis

## Abstract

Previously established diagnostic approaches for the diagnosis of bacterial vaginosis (BV), such as the Amsel criteria or the Nugent scoring system, do not always correspond to modern trends in understanding the etiology and pathogenesis of polymicrobial conditions. Inter-examiner variability and interpretation of data complicate the wet mount microscopy method. Gram staining of smears does not always provide reliable information regarding bacterial taxa, biofilms, or vaginal dysbiosis. Therefore, the introduction of molecular techniques into clinical practice is extremely relevant. Molecular approaches allow not only the diagnosis of BV but also provide an assessment of microbial composition, which is especially important in the differential diagnosis of vaginal infections. The current review represents an expert opinion on BV diagnosis and is based on extensive experience in the field of vaginal infection diagnosis and treatment.

## 1. Introduction

Available clinical guidelines for the diagnosis of bacterial vaginosis (BV) suggest using classical microscopic methods, such as vaginal smear screening (as a part of the Amsel criteria) or Gram staining of smears (as a part of the Nugent criteria). However, it should be highlighted that both methods result in an approximate assessment of the vaginal microbial composition. Our test for the assessment of vaginal microbiota determines the cellular composition, including the presence of “clue” and “pseudo clue” cells, the lactobacilli species, as well as the presence of basal and parabasal cells as a result of epithelial desquamation, yeast-like fungi, and trichomonas. The leukocytes/squamous cells ratio, the presence of basal and parabasal cells as well as “clue” and “pseudo clue” cells, the detected lactobacilli species, and other microorganisms are assessed in stained smears. At the same time, attention is paid to an increased number of epithelial cells heavily covered with adherent bacteria (“clue” cells) and decreased leukocyte number (leukocytes/epithelial cells ratio < 1:1) [1].

The methods combining direct visualization of bacteria and indicative of their cell wall structure are rapid and cost-effective. However, it should be noted that the microscopic assessment of the stained smears depends on the skills and expertise of the researcher, and reliable information regarding the bacterial taxa, biofilms, and/or vaginal dysbiosis is not always obtained.

Culture-based approaches are time-consuming. However, sometimes bacteria cannot be identified, especially difficult-to-culture bacterial taxa. In addition, these approaches do not identify the exact composition of mixed bacterial communities or other infections, which is important in BV [2].

Recently, molecular testing techniques such as real-time polymerase chain reaction (RT-PCR) and multiplex next-generation sequencing (NGS) are gaining increasing use in clinical practice as they allow quantitative detection and accurate identification of bacteria, including those associated with BV. A multiplex PCR test follows an established sequencing algorithm for BV detection and identifies *Gardnerella* spp. and *Fannyhessea vaginae* (formerly known as *Atopobium vaginae)*, as well as lactobacilli species and other BV-associated microorganisms [3]. Another technique employed in the molecular diagnosis of BV is fluorescence in situ hybridization (FISH), simultaneously assessing bacterial taxa composition and their spatial arrangement.

Table 1 contains data on methods for BV diagnosis and their main characteristics [4].

## 2. Multiplex Real-Time Polymerase Chain Reaction (RT-PCR)

Currently, nucleic acid amplification tests (NAAT) are widely used in the diagnosis of vaginal disorders and complement “classic” laboratory methods such as cultures and microscopic examination. They have a number of advantages over routine microbiological studies, such as being able to identify a wide range of microorganisms, including bacteria that are difficult to culture and anaerobic bacteria, viruses, and protozoa. In addition, they are able to determine the number and ratio of microorganisms in the total bacterial mass.

Nucleic acid amplification tests, such as PCR, are theoretically believed to detect only one microorganism in a sample. Numerous studies and data regarding commercial NAAT tests have been published. The technique is based on exponential enzymatic multiplication of a specific nucleic acid sequence, resulting in the production of billions of sequence copies in a short period of time. The amplified product is then easily detected by DNA probes.

The molecular method used in the diagnosis of BV is a specific quantitative real-time PCR (RT-PCR) test. It is a quantitative, reproducible, and reliable molecular tool that determines which bacteria are present in the vaginal biotope of BV-affected women, such as *Atopobium vaginae* (*Fannyhessea vaginae*), *BVAB2*, *Gardnerella vaginalis*, *Leptotrichia/Sneathia* spp., *Megasphaera* spp., and *Mobiluncus* spp. [13].

Multiplex PCR is a type of real-time PCR with fluorescent-labeled probes that allows several PCR tests to be carried out simultaneously in one tube, detecting several pathogens at a time.

The RT-PCR test is abundantly used because DNA amplification can be observed in real-time, eliminating the need for post-amplification analysis and reducing the likelihood of contamination.

Considering the polymicrobial nature of BV, it is desirable to amplify more than one target sequence at a time. Consequently, quantitative multiplex PCR assays are becoming the spotlight of research. Multiplex PCR includes unique sets of primers and probes that bind 16S rRNA gene regions in order to provide a rapid and simple alternative to methods estimating gene copy numbers or expression levels. Different bacterial species associated with BV, when tested alone, have variable positive predictive values in BV diagnosis. However, the combined detection of several bacterial species improves and enhances the test’s performance.

A number of commercial molecular diagnostic tests for BV diagnosis are available, including NuSwab R multiplex PCR [14], the SureSwab BV real-time DNA quantitative PCR assay, the BD Max vaginal panel, and the multiplex BV assay [15]. Thus, these methods establish the diagnosis of BV with a sensitivity of 90.5% to 96.7% and a specificity of 85.8% to 95% when compared to Amsel’s criteria and Nugent’s system [16].

In the Russian Federation, several multiplex real-time PCR tests have been developed and registered. For instance, the Femoflor-16 test is designed to detect the DNA of opportunistic bacteria, lactobacilli, and human DNA (as a sampling control). The total bacterial DNA is determined and vaginal microbiota is assessed on the basis of the ratio of different species—normal vaginal microbiota or dysbiosis. Dysbiosis, in turn, is categorized by its severity (moderate or severe dysbiosis) and by the predominance of aerobic or anaerobic microorganisms (aerobic or anaerobic dysbiosis, respectively). Algorithms for interpreting the Femoflor-16 results do not include the BV category. However, the study conducted by Nazarova V.V. et al. (2017) demonstrated that the manufacturer’s criteria for the Femoflor-16 test in terms of severe anaerobic dysbiosis reflect the characteristics of vaginal microbiota seen in BV. According to their results, the test determines BV (severe anaerobic dysbiosis) with a sensitivity of 99% and a specificity of 93% [17].

At present, a new method is actively being introduced into practice. This test is based on a quantitative assessment of the total vaginal bacterial count (TBC) using the multiplex REAL-TIME PCR Detection Kit—Femoflor. A quantitative analysis of TBC and genius/species-specific DNA of *Lactobacillus* spp., (*L. crispatus*, *L. iners*, *L. gasseri* и *L. jensenii*, and *Bifidobacterium* spp.), and facultative anaerobic microorganisms (*Staphylococcus* spp., *Streptococcus* spp., Enterobacterales, *Enterococcus* spp., and *Haemophilus* spp.), obligate anaerobic microorganisms (*Gardnerella vaginalis*, *Mobiluncus* spp., *Atopobium vaginae* (*Fannyhessea vaginae*), *Anaerococcus* spp., *Bacteroides* spp./*Porphyromonas* spp./*Prevotella* spp., *Sneathia* spp./*Leptotrihia* spp./*Fusobacterium* spp., *Megasphaera* spp./*Veilonella* spp./*Dialister* spp., *Clostridium* spp./*Lachnobacterium* spp., *Peptostreptococcus* spp.), as well as mollicutes (*Candida* spp., *Candida albicans*, *Ureaplasma urealyticum.*, *Ureaplasma parvum*, and *Mycoplasma hominis*), STD pathogens (*Mycoplasma genitalium*, *Chlamydia trachomatis*, *Neisseria gonorrhoeae*, and *Trichomonas vaginalis*), and viruses (HSV1, HSV2, CMV, and HPV) is automatically obtained. The advantages of the Femoflor test over NGS are the following: in addition to the evaluation of microorganisms, including lactobacilli and viruses, it also determines the total bacterial mass/bacteria ratio, and, importantly, establishes the diagnosis of BV with a sensitivity of 84.8% and a specificity of 96.2% [18].

Another test, AmpliSens^®^ Florocenosis/Bacterial Vaginosis-FRT, is aimed at detecting the DNA of *Lactobacillus* spp., *G. vaginalis*, and *A. vaginae*, as well as the total DNA of bacteria colonizing the vagina. Currently, it assesses the TBC/lactobacilli/opportunistic BV-associated microorganism (*G. vaginalis* and *A. vaginae*) ratio in the vaginal biotope. According to the data published in 2016, the sensitivity and the specificity of the Florocenosis/BV-FRT test comprise 100% and 91%, respectively, in comparison to Amsel’s criteria [12]. At present, this test is known as the “Floroskirin” kit.

## 3. Fluorescence In Situ Hybridization (FISH)

Bacterial vaginosis (BV) is characterized by a high recurrence rate due to so-called biofilm formation. Currently, biofilm-associated and non-biofilm-associated forms of BV are distinguished. *Fannyhessea vaginae* (*Atopobium vaginae*) and *Prevotella bivia* are known to exert synergistic interactions with *Gardnerella* spp., resulting in BV.

*G. vaginalis*, being a microaerophilic and facultative anaerobic microorganism, tolerates high redox potential created by microbiota predominated by vaginal lactobacilli, allowing them to coexist. After intercourse, virulent strains of *Gardnerella* spp. displace vaginal *Lactobacillus* spp. and form a biofilm on the surface of the vaginal epithelium. Later, *P. bivia* joins the biofilm formed by *Gardnerella* spp. and microorganisms exert synergistic relationships with each other. *Gardnerella*-induced proteolysis results in the production of amino acids enhancing the growth of *P. bivia*; ammonia produced by *P. bivia*, in turn, promotes *Gardnerella* spp. growth. In addition, both bacteria produce an enzyme, sialidase, which destroys the layer of mucin on the vaginal epithelium. Because of the alteration of the protective mucous layer, the adhesion of other BV-associated bacteria, including *F. vaginae*, to the biofilm increases. Currently, the role of other bacteria in the pathogenesis of BV remains unknown and requires further investigation [19]. Thus, polymicrobial biofilms are detected in 90% of BV-affected women [20,21].

The FISH (fluorescence in situ hybridization) assay using 16S rRNA gene sequencing for biofilm detection has been employed, allowing the accurate detection of biofilm-associated and non-biofilm-associated BV. A large number of ribosomal 16S rRNA copies (10^3^–10^5^) characteristic of *Gardnerella* spp., *F. vaginae*, *Bifidobacteriaceae*, lactobacilli species, and other eubacteria is detected. Bacteria are assessed in a multi-color analysis using FISH probes stained with different dyes, thus identifying certain species within a polymicrobial environment. FISH is suitable for the examination of vaginal smears and urine samples, as well as the endometrium, abortion material, or ejaculate [22].

FISH combines the precision values of both molecular genetics and microscopy, making it possible to assess the relationship between bacteria in their natural microenvironment, such as BV-associated biofilm.

Fluorescent in situ hybridization (FISH) is a cytogenetic method for the identification of target microorganisms (bacteria, yeast-like fungi, and protozoa). The resulting complementary binding (hybridization) of short (usually 18–25 base pairs) fluorescent-labeled target oligonucleotide probes with the ribosomal RNA of an intact cell is analyzed under a fluorescent microscope.

Fluorescent dyes are used for FISH imaging. The first-generation fluorochromes include fluorescein derivatives (fluorescein isothiocyanate—FITC) and rhodamine derivatives, (tetramethyl-rhodamine isothiocyanate (TRITC), 5-(-6-)carboxyfluorescein-N-hydroxysuccimideester (FluoX), and aminomethyl coumarin acetate (AMCA)). Modern fluorochromes, such as cyanine dyes Cy3 or Cy5, have a number of advantages over the first-generation fluorescent molecules. Thus, Cy3-labeled probes have both sufficient luminescence intensity and are resistant to discoloration. Cy5 luminescence is used in the case of multi-stained samples but is located in the spectral part that is not captured by human sight, which means that it requires additional image processing. Labeled Cy3 probes are easily combined with FluorX-labeled probes, although they are less sensitive. In microbiological studies, a combination of four fluorochromes is considered most optimal: FITC as it is the most commonly used, the cyanine dyes Cy3 and Cy5 as they are more resistant to fading, and DAPI for contrasting cellular eukaryotic nuclei. Sets of specific filters can be applied simultaneously [23,24].

This method also rapidly determines different microorganisms. For example, *Gardnerella* genotypes, as well as their relation to each other and to epithelial cells, provide knowledge on whether they are in an adhesive or dispersed state.

In other words, this tool rapidly establishes a diagnosis of recurrent biofilm-associated bacterial vaginosis, which requires two-step therapy sometimes followed by mechanical biofilm destruction and antibiotic and probiotic prescription. On the other hand, if the specimen reveals scattered microorganisms and no biofilm or a false biofilm, such as so-called “pseudo-clue” cells characteristic of *L. iners*, the diagnosis of non-biofilm-associated BV is made. The latter, in turn, does not recur and the management will differ tremendously in this case.

The presence of a structured polymicrobial biofilm on the surface of the vaginal epithelium represents an important diagnostic marker of BV [20]. Interestingly, the presence of “clue cells” in the native preparation has been considered one of the clinical criteria for BV diagnosis since the early 1980s [5]. Only thirty years later, after the introduction of FISH technology, it became clear that “clue” cells represent desquamated epithelial cells heavily covered with microorganisms and are a part of the biofilm [25]. It was the application of the FISH technique that promoted the understanding of the spatial structure within these formations. Biofilm has been described as a continuous layer of small, curved rods (*Gardnerella* spp.) tightly attached to the surface of the vaginal epithelium [26].

The above-mentioned studies provided a new perspective on the term “clue” cell as a part of the biofilm. The authors demonstrated heterogeneity among adherent microorganisms of vaginal smears, where “clue” cells were identified under routine microscopy. Based on the bacteria taxonomy, the FISH technique revealed a high degree of variability in the bacterial layer, which is formed in fundamentally different ways. One is characterized by the adhesive growth of *Gardnerella* on the surface of epithelial cells, resulting in the appearance of true “clue” cells, while “pseudo clue” cells are formed from the sedimentary growth of separately located bacterial groups in the vaginal mucus. These groups, the authors suggest, simply envelop epithelial cells in areas of excessive microbial growth. Accordingly, only the characteristic growth of adherent *Gardnerella* conglomerates on the epithelium surface represents biofilm growth, and the authors proposed to call this condition biofilm-associated BV. The second condition is believed to be bacterial vaginosis or dysbiosis. As a result, the authors obtained only 56% of true “clue” cells and they suggest that this discrepancy in the identification of “clue” cells may explain previously observed inconsistencies in clinical trial results regarding sexual transmission of the disease, the severity and frequency of associated complications, and the lack of therapeutic results in women with BV [27].

Recent studies in molecular genetics have shed new light on genetic heterogeneity and taxonomic diversity within the *Gardnerella* genus. At least 13 distinct strains were reported within the taxon formerly known as *G. vaginalis*.

Since 2019, a fundamentally new taxonomic classification has been proposed, establishing at least 13 separate species within the *Gardnerella* genus. Based on whole-genome sequencing, biochemical properties, and matrix-assisted laser desorption/ionization time-of-flight mass spectrometry (MALDI-TOF), three new species, i.e., *G. piotii*, *G. swidsinskii*, and *G. leopoldii*, were described in addition to *G. vaginalis*. The remaining nine species have not yet been named or described, possibly due to a lack of a sufficient basis for their designation.

Although these species may be closely related genetically, only some of them (pathogenic) are associated with BV, and non-pathogenic species are detected in healthy women [25]. Thus, pathogenic *Gardnerella* species are associated with vaginal biofilm formation, while the role of other microorganisms in BV pathogenesis requires further investigation [28].

Results obtained by A. Krysanova (2021) reported that the simultaneous detection of three or more *G. vaginalis* strains (>8 Lg) was associated with a high recurrence rate of BV in affected women. Three or four strains are detected in women with recurrent courses of BV, while a combination of one, two, or four strains is seen in 78% of cases [29].

The detection of separate *G. vaginalis* genotypes does not have much prognostic value in the establishment of recurrent BV. It is the ratio and simultaneous identification of several genotypes at a time, most often the 1st, 2nd, and 4th, in increased quantity that can help in the prediction and diagnosis of recurrent BV, which, in turn, will improve the therapeutical approach. Since biofilms are formed by various bacteria, mainly *G. vaginalis* [30,31], the genetic analysis of these microorganisms will identify biofilm-forming *Gardnerella* spp. At present, identification of BV-associated biofilms in widespread practice is difficult. It is possible to determine “clue” cells in the vaginal discharge as a biofilm marker, but personnel often misinterpret these results. More training in microscopic examination of vaginal smears is required among laboratory staff. The development and introduction of real-time PCR into practice with subsequent identification of different *G. vaginalis* genotypes will improve the diagnosis of recurrent BV.

Thus, currently, approaches for BV diagnosis are undergoing significant changes. More attention is being paid to molecular techniques due to the importance of not only confirming the condition at the time when the patient presents with complaints. An emphasis should be made on treatment and future prognosis: whether the chances for recurrence are high or if it was a single episode of BV.

In addition, it is important to carry out a differential diagnosis of BV with other vaginal infections due to the fact that different conditions cause similar symptoms, especially in cases of intermediate vaginal microbiota detection (according to Nugent’s criteria), which may include vulvovaginal candidiasis (VVC), aerobic vaginitis (AV), and/or cytolytic vaginitis (CV).

Under microscopic examination of a sample of vaginal discharge, lactobacilli are identified as Gram-positive rods. Their abundance is reflected in low Nugent’s scores and they are considered normal vaginal microbiota. However, some species of *L. iners* are not always stained as Gram-positive, rod-shaped bacteria. Sometimes they look like Gram-variable or Gram-negative curved rods, which is characterized by a thin layer of peptidoglycan within the cell wall [32]. Considering that *L. iners* is dominant in most common vaginal microbiota types, the assessment of vaginal discharge according to Nugent’s criteria is associated with an increased risk of obtaining an unreliable result [32]. In such cases, the application of molecular testing is crucial in order to obtain an accurate laboratory report.

## 4. The Application of Molecular Testing in the Differential Diagnosis of BV and Other Inflammatory and Non-Inflammatory Conditions of the Vagina

The development of diagnostic methods, including 16S ribosomal RNA gene sequencing, expanded our understanding of the role of lactobacilli in the vaginal microbiome and allowed us to determine the features of their metabolism. About 20 species of lactobacilli were discovered within the vaginal biotope, with *L. crispatus*, *L. gasseri*, *L. iners*, and *L. jensenii* being the most common.

The microbiome investigation of healthy reproductive-aged women conducted by Ravel J. et al. (2011) identified five main community state types (CST). Four of them are dominated by *L. crispatus*, *L. gasseri*, *L. iners*, and *L. jensenii*, respectively, and one is characterized by a decreased content of *Lactobacillus* spp. and the dominance of *Sneatia*, *Prevotella*, *Megasphera*, and *Streptococcus* species [33]. CST-I (dominated by *L. crispatus*) and CST-V (dominated by *L. jensenii*) are considered stable and provide low vaginal pH. *L. gasseri* is dominant in CST-II, which is not stable and sometimes transitions to CST-I. CST-III (dominated by *L. iners*) is considered a transitional type and is characterized by a higher pH. CST-IV vaginal microbiota lacks lactobacilli and is divided into two subgroups: CST-IVA, which contains *L. iners* and BV-associated bacteria (BVAB), and CST-IVB with a predominance of BVAB, i.e., *Atopobium*, *Gardnerella*, *Sneatia*, and *Mobiluncus*. This type of vaginal microbiota is most often found in BV-affected women and healthy women of the African ethnic group and is characterized by low lactic acid production and a higher susceptibility to sexually transmitted diseases, pelvic inflammatory disease, preterm labor, and miscarriage. The predominance of *L. iners* and BVAB is associated with a higher content of pro-inflammatory cytokines, such as interleukin-1α, interleukin-18, and tumor necrosis factor-alpha (TNF-α), which are responsible for inducing an inflammatory response in the lower genital tract [34].

When investigating vaginal lactobacilli in 135 healthy women with multiplex PCR, Brolazo et al. (2011) noted that *L. crispatus* (32.6%), *L. jensenii* (25%), and *L. gasseri* (20.6%) were the most commonly detected species [35]. In 2014, data on the development and validation of a new TaqMan quantitative polymerase chain reaction for the identification and quantification of four major vaginal lactobacilli species, i.e., *L. crispatus*, *L. jensenii*, *L. gasseri*, and *L. iners*, were presented. It has been demonstrated that *L. crispatus*, *L. jensenii* and, to a lesser extent, *L. gasseri* are common in healthy women, while the dominance of *L. iners* is associated with BV [36].

Our studies determined the prevalence rate of various lactobacilli species in women with different vaginal microbiota. In the population of reproductively aged women, the most frequent dominant species are *L. iners*, *L. crispatus*, *L. gasseri*, and *L. jensenii*, which necessitates the differentiation and quantification of these four species. The vaginal biotope of reproductively aged women is colonized, as a rule, by several lactobacilli species; however, only one or, rarely, two species are dominant. *Lactobacillus iners* is the most common dominant microorganism in the population of Russian women in St. Petersburg. The dominance of *Lactobacillus crispatus* is more often observed in women with normal vaginal microbiota and is not detected in women with bacterial vaginosis and mixed vaginal infections. A vaginal biotope predominated by *L. crispatus* is considered the most stable; the dominance of *L. iners* and *L. gasseri* is associated with the risks of vaginal dysbiosis. The prevalence of *L. crispatus* among women with normal vaginal microbiota was reliably higher when compared to women with BV and anaerobic vaginitis (*p* < 0.0001). The predominance of *L. crispatus* is known to be associated with lower vaginal pH and negatively correlates with *Gardnerella vaginalis*/*Prevotella bivia*/*Porphyromonas* spp. (*p* < 0.0001), which allows us to consider the dominance of *L. crispatus* in the vaginal microbiota as a reliable marker of BV protection.

Currently, the “Lactobacilli Typing” test for the identification of the total lactobacilli count in addition to the most common vaginal lactobacilli species (*L. crispatus*, *L. iners*, *L. jensenii*, *L. gasseri*, *L. johnsonii*, *L. vaginalis*, and *L. acidophilus*) has been introduced into practice [37].

Vulvovaginal candidiasis (VVC) is a common, often recurrent condition affecting millions of women worldwide and is caused primarily by *Candida albicans* and, to a lesser extent, by other species, including two closely related pathogens, *Candida africana* and *Candida dubliniensis*. With a detection rate of *C. albicans* in VVC of 67%, deeper identification demonstrated that in 42% of cases, these were yeast-like fungi *C. dubliniensis* [38].

Based on the results of NAAT, 119 samples with *C. albicans* (89.47%), 11 with *C. africana* (8.27%) and three with *C. dubliniensis* (2.26%) were identified among 133 women with VVC [39].

For the purpose of differentiating between BV and other inflammatory and/or non-inflammatory vaginal conditions, molecular tests identifying lactobacilli species (such as *L. crispatus*, *L. iners*, *L. jensenii*, *L. gasseri*, *L. johnsonii*, *L. vaginalis*, and *L. acidophilus*) and fungal agents (*Candida*, *Malassezia*, *Saccharomyces*, and *Debaryomyces*) should be used in routine practice.

At our center, the real-time PCR MicozoScrin test is used for the identification of fungal infections caused by *Candida*, *Malassezia*, *Saccharomyces*, and *Debaryomyces: Meyerozyma guilliermondii* (*C. guilliermondii*), *Candida albicans, Pichia kudriavzevii* (*C. krusei*), *Saccharomyces cerevisiae, Candida auris, Candida tropicalis, Clavispora lusitaniae* (*C. lusitaniae*), *Debaryomyces hansenii* (*C. famata*), *Candida dubliniensis, Candida glabrata, Candida parapsilosis, Malassezia* spp., *Kluyveromy cesmarxianus* (*C. kefyr*), and *Malassezia furfur* [40].

In order to highlight the importance of molecular techniques in the differential diagnosis of BV, clinical case reports are provided.
1.A 37 y.o. patient diagnosed with vulvovaginal candidiasis is planning to conceive. According to Nugent’s criteria, an intermediate vaginal microbiota (4 points) was identified. The assessment according to Savicheva’s criteria revealed the leukocytes/epithelial cells ratio to be 5:1, abundant lactobacilli, budding yeast cells, and no hyphae of yeast-like fungi. What was the causative agent in this particular case? A real-time PCR “Femoflor” test reported the presence of lactobacilli (6.1 Lg) and *Candida* spp. (4.3 Lg). Simultaneously, the MicozoScrin test was conducted and *C. krusei* (1.9 Lg), *Saccharomyces cerevisiae* (3.3 Lg), *C. glabrata* (4.8 Lg), and *C. kefir* (3.2 Lg) were detected. The “Lactobacilli Typing” test was carried out on the same patient, resulting in the presence of *L. iners*.

Thus, the complex molecular approach in this patient allowed for the confirmation of *Candida*-non-*albicans* vulvovaginal candidiasis and subsequent treatment prescription. Communities dominated by *Lactobacillus iners*, in contrast to *Lactobacillus crispatus*, are known to occur more frequently with *Candida*. In vitro, *Lactobacillus crispatus* inhibits *Candida* more efficiently than *Lactobacillus iners* due to higher lactic acid production. Therefore, lactobacilli differentiation is important in cases of vulvovaginal candidiasis [41].
2.A 35 y.o. patient wishing to conceive presents complaining of vaginal discharge, vaginal burning, and with a vaginal pH of 3.5. Normal vaginal microbiota was noted (2 points according to the Nugent score). The assessment according to Savicheva’s criteria revealed the leukocytes/epithelial cells ratio to be 8:1, abundant lactobacilli, signs of cytolysis, absent yeast-like fungi and trichomonas, and present parabasal epithelial cells. “Femoflor” reported no pathology and the “Lactobacilli Typing” test detected *L. jensenii* (7.8 Lg). The patient was diagnosed with cytolytic vaginitis and prescribed appropriate treatment.

Thus, the development of cytolytic vaginitis is possible in communities dominated by *L. jensenii*, since this particular microorganism maintains a low pH due to the high amounts of lactic acid produced.

Women with normal vaginal microbiota (Nugent 0–3) may complain of discharge, burning, and itching. The main causative agents of such discomfort are lactobacilli present in the vaginal discharge. The condition characterized by an excessive number of lactobacilli with or without concomitant cytolysis is called lactobacillosis or cytolytic vaginitis (CV)—a debated diagnosis among researchers [42]. Microscopic investigation reveals abundant lactobacilli and evidence of cytolysis. A connection between CV and *L. crispatus* overgrowth has been demonstrated [43]. Clinical manifestation of CV is often similar to that of vulvovaginal candidiasis [44]. Thus, both conditions certainly require accurate molecular testing based not solely on clinical criteria and microscopic findings.
3.A 37 y.o. patient presented with frequent exacerbation of vulvovaginal candidiasis for the past several years. According to Nugent’s criteria, an intermediate vaginal microbiota (5 points) was identified. The assessment based on Savicheva’s criteria revealed the leukocytes/epithelial cells ratio to be 15:1, a reduced number of lactobacilli, present Gram-positive cocci, budding yeast cells, and hyphae. “Femoflor” confirmed the absence of lactobacilli, *Streptococcus* spp. (8.5 Lg), and *Candida* spp. (5.8 Lg). *Candida albicans* was identified with a “MicozoScrin” test. The patient was diagnosed with mixed vaginitis (anaerobic vaginitis and vulvivaginal candidiasis) and prescribed appropriate treatment.

Thus, it is necessary to use available molecular tools for the diagnosis of vaginal infections, considering their mixed nature.

## 5. Local Immune Response Evaluation

Normal vaginal microbiota is maintained, on one hand, by the dominance of lactobacilli in the vaginal biotope and, on the other hand, by complex synergetic interactions between secretory proteins, peptides, epithelial cells, and immune cells. All the components of this system should be in a certain balance in order to maintain the health of the vaginal mucosa. A disruption in each step may lead to the development of an infectious process [45]. A significant role in this process belongs to pro- and anti-inflammatory cytokines involving epithelial, endothelial, and dendritic cells. Even minor alterations in the composition of the vaginal microbiota result in a local immune response, causing changes in immune molecule levels produced by macrophages, dendritic cells, neutrophils, and natural killer (NK) cells, as well as vaginal epithelial cells. Phagocytosis and the enzymatic cleavage of pathogens are accompanied by pro-inflammatory cytokine production (TNF-α, IL1ß, IL-6, etc.), which in turn attract immune cells to the site of inflammation [46].

Therefore, in order to assess the markers of a local inflammatory response, the Immuno Quantex molecular test was developed which determines IL-1b, IL-10, IL-18, TNFα, TLR4, GATA3, CD68, and B2M levels. Significant differences in cytokines’ mRNA expression were revealed in women with vaginal infections compared to healthy controls by O. Budilovskaya (2020). Bacterial vaginosis was characterized by reduced IL-18 and GATA3 mRNA expression; anaerobic vaginitis was accompanied by increased IL-1b, IL-10, and TLR4 mRNA levels; increased IL-1b and TLR4 mRNAs were detected in vulvovaginal candidiasis. Thus, immune markers can be used for the differential diagnosis of vaginal infections. For instance, increased expression of IL-18, GATA3, and CD68 mRNAs was demonstrated in samples with *L. crispatus* dominance. The combination of microbiological and immunological markers represents a favorable prognostic tool in terms of vaginal microbiota stability of the physiological microbiocenosis of the vagina [37].

## 6. Conclusions

In summary, significant progress has been made regarding molecular approaches. At present, it is not enough to use solely clinical or microscopic methods for BV diagnosis. Studies developing molecular techniques for the diagnosis of bacterial vaginosis or assessing vaginal microbiota are emerging. At present, the diagnosis of BV, in particular the recurrent forms, requires the application of multiple diagnostic tools such as microscopic and molecular methods, including FISH which determines biofilm-associated vaginosis, as well as real-time PCR which provides a qualitative assessment of bacteria and their taxonomic identification. Researchers evaluate the vaginal microbiota in different patients: pregnant and non-pregnant women, reproductively aged patients, women in peri- or post-menopause, and infertile patients. Accurate and rapid tests are important for early diagnosis, treatment initiation of significant infections, and ascending infection prevention. The latter, in turn, represents the key to solving a demographic problem.

## Figures and Tables

**Table 1 ijms-25-00449-t001:** Methods for BV diagnosis.

Method	Main Criteria	Advantages	Disadvantages	Characteristics
**Clinical methods**				
Amsel’s criteria [5]	“Wet mount” microscopy Abnormal vaginal discharge with «fishy» odor; Vaginal pH > 4.5; Positive whiff test when vaginal fluid is exposed to 10% potassium hydroxide; “Clue” cells on wet mount.	Onsite diagnostic; symptomatic BV is diagnosed clinically when ≥3 of 4 clinical signs are present	Time-consuming; Biased; Abnormal vaginal discharge is present only in 50% of BV-affected women; Absence of pH strips; Absence of potassium hydroxide; Healthcare providers do not distinguish “clue” cells and “pseudo-clue” cells.	Sensitivity: 37–70%, Specificity: 94–99% when compared to Nugent’s method
Nugent’s system [6]	Microscopic method based on gram-stained smears is calculated by assessing the presence of large Gram-positive rods (*Lactobacillus* morphotypes; scored as 0 to 4), small Gram-variable rods (*Gardnerella vaginalis* morphotypes; scored as 0 to 4), and curved Gram-variable rods (*Mobiluncus* spp. morphotypes; scored as 0 to 2): 0–3—BV negative 4–6—intermediate 7–10—BV positive	Unbiased, cost-effective, simple to perform, 7 + is considered indicative of BV	Does not involve “clue” cells; Delayed result transmission; Time-consuming; “Intermediate” result (4–6) is hard to interpret (possibly vaginal candidiasis, aerobic vaginitis, etc.); Determination of only certain bacterial morphotypes, approximate conclusion on vaginal biotope composition. “Clue” cells are not determined	Sensitivity: 89%, Specificity: 83% when compared to Amsel’s criteria
Hay-Ison’s criteria [7]	Microscopic method based on gram-stained smears and assessing bacterial morphotypes: 0—no bacteria 1—absent BV signs 2—no obvious BV signs 3—BV signs 4—Gram-positive cocci	Unbiased, simple to perform	Delayed result transmission; Does not involve “clue” cells; Time-consuming; “Intermediate” result is hard to interpret; *Lactobacillus* morphotypes are not determined; Determination of only certain bacterial morphotypes, approximate conclusion on vaginal biotope composition.	Sensitivity: 98%, Specificity: 96% when compared to Amsel’s criteria. Positive predictive value: 94% Negative predictive value: 96%
Savicheva’s criteria [8]	Microscopic investigation of vaginal smears for BV signs: Leukocytes/epithelial cells ratio less than 1:1; “Clue” cells; Bacterial morphotype of lactobacilli is low or absent; Other microorganisms are present.	Unbiased; Cells composition determination: “clue” and “pseudo-clue” cells, morthotypes of lactobacilli, basal/parabasal epithelial cells; Inflammatory response evaluation: leukocytes/epithelial cells ratio; Ratio of lactobacilli and other bacteria (dominant, low or absent); Time-effective.	Assessment of bacterial morphotypes.	Sensitivity: 98%, Specificity: 96% when compared to Amsel’s criteria.
**Molecular methods**				
Fluorescence in situ hybridization (FISH) [9]	Fluorescent microscopy using 16S rRNA stained probes	In-situ biofilm/In-situ dysbiosis detection	Cost-consuming; Expensive equipment; Manual test; Selection of specific primers; Needs wider implementation to the practice.	Sensitivity: 84.6%, Specificity: 97.6–100% when compared to Nugent’s method
Next generation sequencing (NGS) [10,11]	16S rRNA gene sequencing	A quantitative assessment of vaginal microbiome	Cost-consuming; Expensive equipment. Results need to be interpreted correctly.	Sensitivity: 95% when compared to clinical methods
Multiplex PCR [3,12]	Quantitative multiplex PCR for BV diagnosis	Commercial automated tests, easy to perform, automated result acquisition.	Cost-consuming; Limited data; No comparison between biofilm detection (FISH) and quantitative PCR.	Sensitivity: 91–97%, Specificity: 77–91% when compared to clinical methods

## Data Availability

Not applicable.

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
