# Peer review of "Molecular Testing for the Diagnosis of Bacterial Vaginosis"

_ijms, 2023, doi:10.3390/ijms25010449_

Round 1

Reviewer 1 Report

Comments and Suggestions for Authors

The authors reviewed the advantages and disadvantages of the existing methods for diagnosing bacterial vaginosis. Also, they reviewed new methods using molecular approaches such as FISH, NGS, and multiplex PCR-based techniques. The authors indicated the importance of combining such molecular approaches and conventional methods to diagnose BV accurately for the following treatments with some case reports. The reviewer believes that the article should provide beneficial information for clinicians.

Comments to the authors

#1. What are the MicozoScrin test (lines 307 and 359) and the MycozoScreen test (line 322)? Is that the new technique developed by the authors? Please indicate the method in detail.

#2. The reviewer recommends that the authors add the multiplex PCR-based method's advantages, especially developed by the authors, over the NGS-based microbiome analysis method.

Author Response

The authors reviewed the advantages and disadvantages of the existing methods for diagnosing bacterial vaginosis. Also, they reviewed new methods using molecular approaches such as FISH, NGS, and multiplex PCR-based techniques. The authors indicated the importance of combining such molecular approaches and conventional methods to diagnose BV accurately for the following treatments with some case reports. The reviewer believes that the article should provide beneficial information for clinicians.

Comments to the authors

Q 1. What are the MicozoScrin test (lines 307 and 359) and the MycozoScreen test (line 322)? Is that the new technique developed by the authors? Please indicate the method in detail.

A 1. The spelling of MicozoScrin was corrected (line 351). This is a molecular real-time PCR test for the diagnosis of fungal infections (detection and typing of Candida, Malassezia, Saccharomyces and Debaryomyces) developed by the DNA-Technology company, Russia. Full description can be found on lines 337-342. In the text, the fungi species are highlighted in italics.

Q 2. The reviewer recommends that the authors add the multiplex PCR-based method’s advantages, especially developed by the authors, over the NGS-based microbiome analysis method.

A 2. The information was added (line 119).

The advantages of the Femoflor test over NGS are the following: in addition to the evaluation microorganisms, including lactobacilli and viruses, it also determines the total bacterial mass/bacteria ratio, and, importantly, establishes the diagnosis of BV with sensitivity 84.8% and specificity 96.2%. [18]

Reviewer 2 Report

Comments and Suggestions for Authors

The review by Savicheva concerns an important topic, the molecular diagnosis of infections. Of particular relevance, the review is about bacterial vaginosis, a recurrent infection of difficult diagnosis. The review is well-written and provides examples of clinical cases to strengthen the suggestions/conclusions of the author. I have only one smal suggestion:

Please provide references for the "Lactobacilli Typing", "Real-time PCR MicozoScrin test", "real-time PCR “Femoflor” test"

Author Response

The review by Savicheva concerns an important topic, the molecular diagnosis of infections. Of particular relevance, the review is about bacterial vaginosis, a recurrent infection of difficult diagnosis. The review is well-written and provides examples of clinical cases to strengthen the suggestions/conclusions of the author. I have only one smal suggestion: Please provide references for the “Lactobacilli Typinп”, “Real-time PCR MicozoScrin test”, “real-time PCR “Femoflor” test”

“Lactobacilli Typing” – reference number 38

“Real-time PCR MicozoScrin test” - reference number 41

real-time PCR “Femoflor” test - reference number 18

Reviewer 3 Report

Comments and Suggestions for Authors

The manuscript, entitled “The Importance of Molecular Testing for the Diagnosis of Bacterial Vaginosis. An Expert Opinion” described that previously established diagnostic approaches, such as Amsel criteria or Nugent scoring system, do not always correspond to modern trends in understanding the etiology and pathogenesis of polymicrobial condition. The authors concluded several molecular techniques, which allow not only diagnosing BV but also provide assessment of microbial composition. However, due to some drawbacks, my suggestion is major revision.

Comments:

1. In table 1, how did the authors calculate the sensitivity and specificity of each method? What are the criteria? What’s more, it is suggested to supplement the references of each method.

2. Why did the authors highlight Gardnerella strains in the section of 4. Genome investigation of Gardnerella strains? The title of this review is The Importance of Molecular Testing for the Diagnosis of Bacterial Vaginosis. An Expert Opinion, but why emphasized the strains? Not only Gardnerella strains can be investigated by genome, the strains which can be cultured can be investigated by genome. So, the subtitle of section 4 is not appreciated.

3. From table 1 we can see that three molecular methods have disadvantage that is cost-consuming. Compared with traditional methods, the advantages of molecular methods are not significant. And Gynecologists still prefer traditional testing methods in hospital now.

4. It is suggested to add the detection principle of molecular methods in the section of 2, 3, and 4.

5. In addition, except for three molecular methods mentioned in this study, are there any other molecular methods? It should be comprehensive to introduce all key methods.

6. There are some format errors in the text. For example, the format of references is not consisitent.

Comments on the Quality of English Language

Moderate editing of English language required.

Author Response

The manuscript, entitled “The Importance of Molecular Testing for the Diagnosis of Bacterial Vaginosis. An Expert Opinion” described that previously established diagnostic approaches, such as Amsel criteria or Nugent scoring system, do not always correspond to modern trends in understanding the etiology and pathogenesis of polymicrobial condition. The authors concluded several molecular techniques, which allow not only diagnosing BV but also provide assessment of microbial composition. However, due to some drawbacks, my suggestion is major revision.

Comments:

  1. In table 1, how did the authors calculate the sensitivity and specificity of each method? What are the criteria? What’s more, it is suggested to supplement the references of each method.

A1. References to each method were added to the table. Both indicators were mentioned on the basis of previously published articles. The list of references was also revised.

  1. Why did the authors highlight Gardnerella strains in the section of 4. Genome investigation of Gardnerella strains? The title of this review is The Importance of Molecular Testing for the Diagnosis of Bacterial Vaginosis. An Expert Opinion, but why emphasized the strains? Not only Gardnerella strains can be investigated by genome, the strains which can be cultured can be investigated by genome. So, the subtitle of section 4 is not appreciated.

A2. I agree with the reviewer. Subtitle 4 was removed. However, I suggest keeping genome investigation of Gardnerella strains as I consider it a new step towards the diagnosis of recurrent BV (when several genotypes are detected simultaneously), its prognosis and treatment approach.  

  1. From table 1 we can see that three molecular methods have disadvantage that is cost-consuming. Compared with traditional methods, the advantages of molecular methods are not significant. And Gynecologists still prefer traditional testing methods in hospital now.

A3. I agree that Amsel and microscopic methods are more cost-effective than molecular methods. However, neither Amsel method nor microscopy are widely used in many countries. When introduced into practice, FISH will be also cost-effective. Molecular methods are more reliable in assessment of vaginal biotope. Moreover, the intermediate vaginal microbiota assessed by the Nugent scale, often implies different forms of vaginal inflammation. The development of molecular methods provides their application in BV diagnosis and/or differential diagnosis with other non-BV related conditions.

  1. It is suggested to add the detection principle of molecular methods in the section of 2, 3, and 4.

A4. The detection principle was added to all sections.

Line 74. Multiplex PCR is a type of real-time PCR with fluorescent-labeled probes allowing several PCR tests to be carried out simultaneously in one tube, detecting several pathogens at a time.

Line 160. Fluorescent in situ hybridization (FISH) is a cytogenetic method for the identification of target microorganisms (bacteria, yeast-like fungi and protozoa). The resulting complementary binding (hybridization) of short, usually ranging 18-25 base pairs of fluorescent-labeled target oligonucleotide probes with ribosomal RNA of an intact cell is analyzed under fluorescent microscope.

Fluorescent dyes are used for FISH imaging. The first generation fluorochromes include: fluorescein derivatives (fluorescein isothiocyanate - FITC), rhodamine derivatives (tetramethyl-rhodamine isothiocyanate - TRITC), 5-(-6-) carboxyfluorescein-N-hydroxysuccimideester (FluoX) and aminomethyl coumarin acetate (AMCA). Modern fluorochromes, such as cyanine dyes Cy3 or Cy5, have a number of advantages over the first generation fluorescent molecules. Thus, Cy3 labeled probes, have both sufficient luminescence intensity and are resistance to discoloration. The Cy5 luminescence is used in case of multi-stained samples, but is located in the spectral part that is not captured by the human sight, which means it requires additional image processing. Labeled Cy3 probes are easily combined with FluorX-labeled probes, although they are less sensitive [23]. In microbiological studies, the combination of four fluorochromes is considered most optimal: FITC being the most commonly used, cyanine dyes Cy3 and Cy5, as more resistant to fading and DAPI - for contrasting cellular eukaryotic nuclei. Using sets of specific filters, they can be applied simultaneously [24].

  1. In addition, except for three molecular methods mentioned in this study, are there any other molecular methods? It should be comprehensive to introduce all key methods.

A5. Currently, only these three molecular tests are used for BV diagnosis. For sure, the diagnosis of BV must be comprehensive. The combination of microscopic method and real-time PCR are most abundantly used. This is highlighted in conclusion (line 423).

  1. There are some format errors in the text. For example, the format of references is not consisitent.

A6. Corrected.

Round 2

Reviewer 3 Report

Comments and Suggestions for Authors

The author of the manuscript, entitled “The Importance of Molecular Testing for the Diagnosis of Bacterial Vaginosis. An Expert Opinion” has revised the text according to the comments. However, due to some drawbacks, my suggestion is minor revision.

Comments:

1.     The full names of some abbreviations, like PPV and NPV, should be provided.

2.     The content of table 1 is concise, especially about advantages and disadvantages. It is suggested to expand them.

3.     English writing is recommended to be polished.

Comments on the Quality of English Language

Moderate editing of English language required.

Author Response

The author of the manuscript, entitled “The Importance of Molecular Testing for the Diagnosis of Bacterial Vaginosis. An Expert Opinion” has revised the text according to the comments. However, due to some drawbacks, my suggestion is minor revision.

Comments:

  1. The full names of some abbreviations, like PPV and NPV, should be provided.

А. Corrected

  1. The content of table 1 is concise, especially about advantages and disadvantages. It is suggested to expand them.

А. Corrected

  1. English writing is recommended to be polished - revised.
